

# Land use change associated with urbanization modifies soil nitrogen cycling and increases N₂O emissions

**Lona van Delden, David W Rowlings, Clemens Scheer, Peter R Grace**

Queensland University of Technology (QUT), Brisbane, QLD, Australia

5   *Correspondence to*: Lona van Delden (lona.vandelden@qut.edu.au)

**Abstract.** Urbanization is becoming increasingly important in terms of climate change and ecosystem functionality worldwide. We are only beginning to understand how the processes of urbanization influence ecosystem dynamics, making peri-urban environments more vulnerable to nutrient losses. Brisbane in South East Queensland has the most extensive urban sprawl of all Australian cities. This research estimates the environmental impact of land use change associated with urbanisation by examining soil nitrogen (N) turnover and subsequent nitrous oxide ($N_2O$) emissions with a fully automated system that measured emissions on a sub-daily basis. There was no significant difference in soil $N_2O$ emissions between a native dry sclerophyll eucalypt forest and an extensively grazed pasture, wherefrom only low annual emissions were observed amounting to 0.1 and. 0.2 kg $N_2O$ ha$^{-1}$ y$^{-1}$, respectively. The establishment of a fertilised turf grass lawn increased soil $N_2O$ emissions by 18 fold (1.8 kg $N_2O$ ha$^{-1}$ y$^{-1}$) with highest emission occurring in the first 2 month after establishment. Once established, the turf grass lawn presented relatively low $N_2O$ emissions after fertilization and rain events for the rest of the year. Soil moisture was significantly higher and mineralised N accumulated in fallow land, resulting in highest $N_2O$ emissions (2.8 kg $N_2O$ ha$^{-1}$ y$^{-1}$) and significant nitrate ($NO_3^-$) losses of up to 63 kg N ha$^{-1}$ from a single rain event due to plant cover removal. The study concludes that urbanization processes into peri-urban ecosystems can greatly modify N cycling and increase the potential for losses in form of $N_2O$ and $NO_3^-$.

## 1 Introduction

Global urbanization processes are becoming increasingly important in terms of global warming and ecosystem functionality. Urban populations worldwide have not only exceeded rural populations but are also predicted to account for all future population growth (United Nations 2008). Urban sprawl and increasing population densities are causing severe land use changes from intact biomes and commercially focused agriculture into smaller residential properties with introduced species. This transition from rural to semi-rural, i.e. peri-urban, and urban environments is increasingly associated with development and construction processes and the extensive establishment of turf grass for residential backyards, public parks and sportsgrounds, and golf courses (IPCC 2006). How these urbanization processes influence ecosystem dynamics in biogeochemical cycling, and therefore contribute to ecosystem vulnerability and global warming is only beginning to be understood.



The consequences of land use change from native to agriculture have been identified by several studies, including a loss in soil quality (structure and nutrient losses) and quantity (erosion), increase greenhouse gas (GHG) emissions, and reduced potential for soil carbon (C) sequestration (Grover et al. 2012; Livesley et al. 2009). On the other hand, changing soils from agricultural to residential use in temperate climates has shown the potential to improve critical ecosystem services by (i) providing stormwater treatment, (ii) acting as a sink for atmospheric nitrogen (N) and (iii) sequestering C (Golubiewski 2006; Raciti et al. 2011).

Studies on the impact of those land use changes on climate change are few but suggest that urbanization will change the biogeochemical cycling of C and N and associated nutrient turnover (Grimm 2008). These biogeochemical alterations induced by land use change interact with urban effects (Betts 2007); such as creating head islands through vegetation replacement and surface sealing but also increased local carbon dioxide concentration of over 500 ppm around cities compared to 390 ppm in natural environments (IPCC 2013; Pataki et al. 2007). These changing local climatic conditions and their feedback effects onto natural ecosystems make peri-urban environments more vulnerable to nutrient losses and potential sources of GHG emissions. With these peri-urban areas expanding worldwide it is most likely that these changing local climates will increasingly have an impact on global climate change, making an examination of GHG emissions from peri-urban land uses all the more urgent.

Nitrous oxide ($N_2O$) along with carbon dioxide ($CO_2$) and methane ($CH_4$) is one of the major greenhouse gases with a global warming potential (GWP) nearly 300 times that of $CO_2$ (IPCC 2013). Nitrous oxide is produced principally by microorganisms during nitrification and denitrification processes from mineral N ($NH_4^+$ and $NO_3^-$) in the soil. The production of $N_2O$ is influenced by a number of soil parameters including substrate availability, temperature, and availability of oxygen, which is dependent on water content and texture of the soil (Rowlings et al. 2015). With the predicted climatic changes, Australia's ancient and fragile soils will most likely be affected in their balance between GHG gas emissions and consumptions (Baldock et al. 2012). Management practices such as fertilization and irrigation enhance $N_2O$ production in the soil by increasing the mineral N content and limiting the oxygen availability (Rowlings et al. 2013; Scheer et al. 2008). Turf grass is the most highly managed land use of peri-urban environments in terms of fertilization, irrigation and frequent mowing, which therefore has a high potential for $N_2O$ emissions.

Research on urban and peri-urban areas in temperate zones suggests that changes in biogeochemical cycling due to urbanization will substantially affect the global climate comparable to agriculture, with those areas and their intensive management expanding rapidly worldwide (Groffman and Pouyat 2009; Milesi et al. 2005). More than half the world's 7.2 billion population currently occupies 2.4 % of the global terrestrial land surface in urban areas (Potere and Schneider 2007; United Nations 2013). While peri-urban environments are often considered too small to be of consequence, the rapid growth of the industry as highlighted in the USA covers over 160,000 $km^2$ occupied with turf grass, three times more than any other irrigated crop in the country (Groffman and Pouyat 2009; Milesi et al. 2005). In Australia about 60 % of all anthropogenic $N_2O$ emissions come from cropped and grazed soils and the first GHG estimations from turf grass establishment support the emission intensity reported from temperate zones (AGO 2010; van Delden et al. 2016). The use of turf grass is consistently



growing in Australia with up to 17,320 ha in turf grass sales and an approximate gross value production of $240 million AUD per annum (ABS 2012; Turf Australia 2012). Detailed estimates of turf grass cover, however, currently do not exist for the Australian continent and other subtropical regions like South-east Asia, China, India or Mexico. Urbanization is currently

neglected in modelled IPCC climate scenarios, mainly due to limited data on C and N processes in urban and peri-urban environments (IPCC 2006, 2013).

Therefore, this study aims to identify the impact of those land use changes associated with urbanization on annual $N_2O$ emissions and their driving parameters in subtropical peri-urban environments. Following a short-term (80 days) GHG sampling campaign focussing on lawn establishment (van Delden et al. 2016), a fully automated closed static chamber system was used to continuously monitor $N_2O$ fluxes together with soil biogeochemical processes over a full year to

determine the seasonal impact of construction work and conversion from extensively grazed pasture to turf grass lawn. This study's high resolution flux measurements and supporting soil N mineralisation illustrates the vulnerability of ecosystems to urbanization processes and the potential impact on N cycling and $N_2O$ emissions.

## 2 Materials and Method

### 2.1 Site description

The study was conducted at the Samford Ecological Research Facility (SERF) in the Samford Valley, 20 km from Brisbane in South-East Queensland, Australia. The Samford Valley covers an area of approximately 82 $km^2$ and is surrounded by mountains to the north, west and south. Mostly cleared in the early 1900s, the valley was developed in the 1960s for dairy and beef cattle as well as intensive agriculture including banana and pineapple. Samford's population density has increased rapidly, almost doubling from 1996 – 2006, causing land use change from predominately rural to residential properties

(Moreton Bay Regional Council 2011). As a result, SERF contains the last remnant forest of the valley floor. The valley is influenced by a humid subtropical climate with seasonal summer rain. The long term mean annual precipitation is 1110 mm with mean annual minimum and maximum temperatures of 13 °C and 25.6 °C respectively (BOM). The soil at the experimental site is characterised by a strong texture contrast between the A and B horizon and is classified as brown Chromosol according to the Australian soil classification (Isbell 2002) and Planosol according to the World Reference Base

(WRB 2015).

### 2.2 Experimental design

This study examines the impact of land use change from a native forest to well-established pasture, turf grass lawn and fallow soil without plant cover using the same sampling campaign setup as van Delden et al. (2016). Each land use treatment included 3 replicated plots, 2 m wide by 10 m long. The turf grass lawn and fallow treatments were established within the

well-established pasture to create a randomised plot design, 50 m from the native forest. The SERF native forest (*Dry sclerophyll eucalypt forest*) is a baseline for historical land use and was unmanaged. The well-established *Chloris gayana*



pasture represents rural development in the area and has been extensively grazed for the last 15 years. Livestock, however, was excluded over the course of the study and the pasture grass was slashed 5 times during the study to ensure it did not exceed the maximum height of the GHG measurement chamber.

The turf grass lawn was established from the well-established pasture by removing 5 cm of topsoil with grass roots. The soil
was rotary hoed twice to a depth of 15 cm and Blue Couch (*Digitaria didactyla*) was planted with 50 kg N ha$^{-1}$ fertilization (13.6.2013) to aid in establishment. Over the experimental year the turf grass lawn was fertilized twice (26.10.13, 6.3.2014) with 50 kg N ha$^{-1}$ and irrigated, in all 150 kg N ha$^{-1}$ y$^{-1}$ of Prolific Blue AN fertilizer (12.0 % nitrogen, 5.2 % phosphorus, 14.1 % potassium, 1.2 % magnesium) with two-thirds of the N in the ammonium form. The turf grass lawn was irrigated to a total of 30 mm during drier months as well as after fertilization. The turf grass was mowed with the clippings removed as
soon as the grass grew to the maximum chamber height, once in spring, twice in summer and twice in autumn, and kept free of weeds manually at all times. Fertilization rates were based on half the local industry practices recommendation. Infrequent mowing represents the normal management for residential properties in this region and is normally in response to increased growth in the wetter and warmer summer months.

The fallow treatment simulated the impact of transitional processes such as construction work and plant cover replacement.
In the fallow treatment, the grass cover was removed and the bare soil was rotary hoed twice to a depth of 15 cm. The fallow treatment was kept free from plant cover over the full experimental year with a non-selective herbicide (Bi-Active 360g/L Glyphosate) and a broad leaf herbicide (Double Time, 340g/l MCPA + 80g/l Dicambra). During the experimental year, high resolution sub-daily N$_2$O flux measurements were combined with mineral N analysis and site specific climate and soil moisture measurements.

**2.3 N$_2$O flux measurements**

Nitrous oxide fluxes were determined from mid-June 2013 to mid-June 2014 using an automated sampling system as detailed by Scheer et al. (2014), extending the turf grass establishment phase documented by van Delden et al. (2016) into a full measurement year. The pneumatically operated 50 cm x 50 cm x 15 cm high static chambers were secured to stainless steel bases, permanently inserted 10 cm into the ground. The chambers were moved each week between two bases per
treatment plot, to minimize the influence of the chamber microclimate. The chambers were connected to an automated sampling system and an in-situ gas chromatograph (SRI GC8610, Torrance, CA, USA) equipped with 63Ni Electron Capture Detector (ECD) for N$_2$O. One replicate chamber from each of the four treatments was closed for one hour, and four headspace gas concentrations measured at 15 minute intervals, followed a known calibration standard (0.5 ppm N$_2$O, Air Liquide, Houston, TX, USA). This process was repeated for the remaining two replicate chambers over a full cycle of three
hours, allowing eight flux measurements to be calculated per day, for each of the 12 chambers.





### 2.4 Auxiliary measurements

Soil samples were taken fortnightly from all replicated treatment plots over the experimental year and divided into 2 depths (0-10 cm, 10-20 cm). $NH_4^+$ and $NO_3^-$ were extracted from the soil using a 1:5 KCl solution with 20 g of fresh soil. The extract was analysed for $NH_4^+$ and $NO_3^-$ with an AQ2+ discrete analyser (SEAL Analytical WI, USA). The net

mineralisation rate was determined from differences in mineral content between sampling dates (Hart et al. 1994). Soil moisture and temperature for each treatment were collected using a TDR probe (HydroSense CD 620 CSA) and a PT100 probe (IMKO Germany). Soil moisture was then converted with the treatment specific bulk density (BD) to water-filled pore space (WFPS). Soil samples were taken for site characterization with a hydraulic soil corer to 1 m depth, air dried and sieved to 2 mm. Particle size analysis for soil texture as well as BD, pH and electrical conductivity (EC) analysis were undertaken

according to Carter and Gregorich (2007). Total C and N content of soil and plant material was determined by dry combustion (CNS-2000, LECO Corporation, St. Joseph, MI, USA) from ground samples.

### 2.5 Flux calculations and statistical analysis

Fluxes were calculated from the slope of the linear increase or decrease of the 4 concentrations measured over the closure time and corrected for chamber temperature and atmospheric pressure, using the procedure outlined by Knowles and Singh

(2003) and Scheer et al. (2014). The linear regression coefficient ($r^2$) was calculated and used as a quality check for each flux measurement. Flux rates were discarded if $r^2$ was < 0.85 for $N_2O$ fluxes (Scheer et al. 2013). Daily fluxes from the automated chambers were calculated by averaging sub-daily measurements for each chamber over the 24 hour period. The detection limit determined for the gas sampling system is ± 1.2 g $N_2O$ ha$^{-1}$ d$^{-1}$. Gaps in the dataset were filled by linear interpolation across missing days.

Statistical analysis was undertaken using SPSS Statistics 21.0 (IBM Corp., Armonk, NY). Non-normal distribution meant all cumulative data were log-transformed for ANOVA analysis using Games-Howell as the post-hoc test. Daily $N_2O$ flux differences between treatments were interpreted by plotting 95 % confidence intervals using R studio. A significant difference of $p < 0.05$ between treatments was assumed in case the confidence intervals of all treatments were not overlapping. A Spearman's rho correlation analysis was used to examine relationships between gas fluxes, soil chemistry,

soil moisture and temperature. The significance value ($p$) is shown for each analysis, as well as the correlation coefficient ($r$) with its significance level ($p < 0.05^*$, $p < 0.01^{**}$).

## 3 Results

### 3.1 Site characteristics

The site received 740 mm of rain during the experimental year, substantially less than the long term average (Table 1). Wet

season rainfall was delayed compared to the historic average, with less than half the rainfall in summer (December to



February) compared to autumn (March to May) (Table 2). Substantial out of season rain also fell in the spring with over 200 mm in November alone. Rainfall was highly episodic, with the highest daily rain event of 108.8 mm in March 2014. The mean annual minimum and maximum temperatures for the experimental year were 16.7 °C and 27.1 °C respectively, and light ground frost occurred twice in August. The turf grass and fallow treatment were established within the pasture and therefore share the same soil profile with its characteristics, except for bulk density in the A1 horizon which changed after the turf grass establishment from 1.4 to 1.2 g cm$^{-3}$. Nutrient removal in turf grass clippings added up to 1.8 t C ha$^{-1}$ y$^{-1}$ and 30 kg N ha$^{-1}$ y$^{-1}$ lost from the system during the experiment year. The turf's biomass production was approximately 6.3 kg C ha$^{-1}$ d$^{-1}$ and 0.13 kg N ha$^{-1}$ d$^{-1}$ in dry matter but varied widely depending on fertilization and available water and increased up to 10.4 kg C ha$^{-1}$ d$^{-1}$.

## 3.2 Environmental parameters

The lowest WFPS during the experimental year was 13 % in the forest, with the highest occurring in the pasture which briefly reached saturation in March 2014 (Figure 3). In all treatments the lowest WFPS occurred in spring and summer with an average of 33 and 32 % respectively together with the highest average maximum daily temperatures of 28 and 30 °C. While the highest seasonal WFPS for all treatments occurred in winter, the maximum WFPS occurred during autumn after the heavy rain in March 2014. The forest had significantly lower WFPS throughout the experimental year than all other treatments ($p < 0.01$, Table 3), while the fallow had significantly higher WFPS ($p < 0.01$). No significant difference in WFPS was observed between pasture and turf grass ($p > 0.05$) although during spring, summer and autumn turf grass had lower minimum and maximum values than the pasture. Fallow had significantly higher and forest significantly lower WFPS than pasture and turf grass ($p < 0.01$) throughout the experimental year.

## 3.3 Temporal variability of mineral nitrogen

Averaged over the experimental year the fallow treatment had the highest $NH_4^+$ and $NO_3^-$ content across 20 cm soil profile, followed by turf grass, pasture and forest (Table 3). These differences in mineral N were significant for all treatments ($p < 0.01$) except between pasture and forest ($p > 0.05$). The 0-10 cm depth had higher average mineral N, $NH_4^+$ and $NO_3^-$ than the 10 – 20 cm depth for all treatments with significant differences between all treatments ($p < 0.01$). In 10 – 20 cm soil depth only the fallow had significantly higher mineral N, $NH_4^+$ and $NO_3^-$ contents ($p < 0.01$). Soil $NH_4^+$ showed relatively little temporal variation and remained consistently above 3 kg $NH_4^+$ ha$^{-1}$ while $NO_3^-$ decreased substantially after rain events and fell below detection limit several times in all treatments but the fallow (Figure 1).

Total mineral N in the forest ranged from 8 to 40.1 kg N ha$^{-1}$ 20 cm$^{-1}$ throughout the year with marginally higher mineral N content from 0-10 cm than 10-20 cm with 9.7 and 8 kg N ha$^{-1}$ respectively. Total mineral N in the pasture ranged from 5.1 to 42.4 kg N ha$^{-1}$ 20 cm$^{-1}$, with a comparable distribution in depth than the 0-10 cm and 10-20 cm forest soil with 10.7 and 7.8 kg N ha$^{-1}$ respectively. Total mineral N in the turf grass soil ranged from 9.1 to 127.6 kg N ha$^{-1}$ 20 cm$^{-1}$. The turf grass had twice as much mineral N in 0-10 cm than 10-20 cm depths with 20.7 and 10.1 kg N ha$^{-1}$ respectively. A short term increase





in both $NH_4^+$ and $NO_3^-$ content in the soil was evident after fertilization in June, October and March, which decreased to the background levels after approximately one month. Total mineral N contents in the fallow soil ranged from 19.7 to 160.7 kg N ha$^{-1}$ 20 cm$^{-1}$ with about 2/3 of the mineral N being located in the upper 10 cm. All main changes in the fallow's total mineral N content were caused by variations in $NO_3^-$ rather than $NH_4^+$. The $NO_3^-$ content increased in the fallow until the

major rain event in March when it dropped from 95.5 to 32.8 kg N ha$^{-1}$ 20 cm$^{-1}$. From the linear increase in mineral N content within the upper 10 cm between January and March 2014 a soil mineralization rate of 0.6 kg N ha$^{-1}$ d$^{-1}$ was estimated.

### 3.4 Temporal variation of $N_2O$ fluxes

Daily $N_2O$ fluxes across all treatments ranged from extended periods close to zero to over 123 g $N_2O$ ha$^{-1}$ d$^{-1}$ from the fallow

with the highest WFPS after heavy rain events (Figure 2). The $N_2O$ fluxes from turf grass were more often significantly different on daily bases than any other treatment with 76 days (21 %) of the experimental year, 80 % of this difference occurred in the first two months after establishment. This was followed by the forest with 65 days (18 %), fallow with 58 days (16 %) and pasture with 29 days (8 %). Daily $N_2O$ fluxes from the forest soil showed no substantial temporal variation throughout the experimental year, with minor emission peaks up to 8.1 g $N_2O$ ha$^{-1}$ d$^{-1}$ after large rain events (> 60 mm) in

November and March. From September until October one of the two bases in one pasture replicate emitted substantially more $N_2O$ than the other replicates, however, the exact cause of this is unknown. Without these spatially variable emissions, the annual flux would have been about 40 % lower and therefore comparable to the forest N loss of 0.09 kg N ha$^{-1}$ y$^{-1}$. During the initial emission peak between June and August the daily average $N_2O$ flux from the turf grass was 24 g $N_2O$ ha$^{-1}$ d$^{-1}$, reaching a maximum of 83 g $N_2O$ ha$^{-1}$ d$^{-1}$. Excluding this initial emission peak, daily $N_2O$ flux averaged 1.2 g $N_2O$ ha$^{-1}$ d$^{-1}$

$^1$. The highest annual $N_2O$ flux was measured in the fallow from only 3 large peaks over 19, 10, and 44 consecutive days after rain events which together accounted for 85 % of the total N losses. Over a third of the significantly high daily $N_2O$ fluxes in the fallow occurred from the heavy rain event in March 2014.

Annual $N_2O$ losses were highest in the fallow and turf grass treatments totalling 1.78 and 1.15 kg N ha$^{-1}$ y$^{-1}$ respectively compared to the pasture and forest losses of 0.15 and 0.09 kg N ha$^{-1}$ y$^{-1}$ ($p < 0.01$, Table 3). About 80 % of the annual $N_2O$

losses in the turf occurred in the first 8 weeks after establishment (Figure 3). Mineral N fertilizer input of 150 kg N ha$^{-1}$ y$^{-1}$ and the yearly $N_2O$-N losses from the turf grass lawn corrected for background emissions (zero N fertilization) from the pasture resulted in an emission factor (EF) of 0.7 % (Kroeze et al. 1997).

### 3.5 Environmental parameters influencing $N_2O$ fluxes

Mineral N contents in the forest and fallow soils were not significantly correlated to $N_2O$ fluxes on a daily basis (Table 4).

However, the linear regression shown in Figure 4 identified a clear increase of $N_2O$ emissions with increasing annual mineral N contents for all treatments. This relationship is supported by the substantial $N_2O$ emissions peaks from the fallow and simultaneous decrease in $NO_3^-$ after the two biggest rain events in November 2013 and March 2014, with WFPS above 70 %.





The separate linear regression for all land uses with plant cover, i.e. forest, pasture, and turf grass, identified an even stronger relationship of mineral N and $N_2O$. Forest and turf grass $N_2O$ fluxes were strongly and fluxes from the pasture and fallow moderately correlated to their WFPS. Temperature was moderate negatively correlated to $N_2O$ fluxes as well as mineral N for pasture and turf grass. In the fallow temperature strongly affected mineral N contents but not $N_2O$ fluxes. Mineral N in
the fallow soil was strongly negative correlated to its WFPS mostly because of the strong negative correlation of $NO_3^-$ with WFPS with $r$ = -0.56**.

## 4 Discussion

This study combines the first high frequency estimates of subtropical $N_2O$ fluxes and annual mineral N cycling from dry sclerophyll forests, unfertilized pastures and turf grass lawns, the most common land uses associated with urban and peri-
urban environments. The lack of high frequency field measurements in urban and peri-urban environments makes accurate assumptions and mitigation strategies difficult. Conventional gas sampling methods most likely result in an over- or underestimation of emissions, as the production and release of $N_2O$ can differ in time (Mosier et al. 1998). This research gap together with the strong temporal variability of subtropical heavy rain events underlines the importance of automated high frequency measurements to capture representative soil-atmosphere gas exchange. The subtropical climatic zone represents an
often neglected area of research, despite the subtropics covering 3.26 M ha in Australia alone, as well as large areas on the North and South American continent, Africa and Asia. Differences between other climates and the humid subtropics are the heavy summer rains leading to high soil moisture and temperatures favourable for high soil microbial activity. Therefore this study identified mineral N content and WFPS as the main parameters driving $N_2O$ production in the soil while studies from temperate zones report temperature as the main driver (Butterbach-Bahl and Kiese 2005; Fest et al. 2009).

### 4.1 Mineral N

Mineralised N in form of $NH_4^+$ and $NO_3^-$ determines the production and loss of N via $N_2O$ and depends on climatic parameters like temperature as well as substrate and oxygen availability. Nutrient mineralisation is often faster in sandy soils but the rapid infiltration of the A horizon of the Chromosol increases the leaching potential of the highly mobile $NO_3^-$ into deeper layers of the soil profile after heavy rain events. The leached $NO_3^-$ is not only lost for plant uptake but can pollute
groundwater and open waterways. In this study mineral N contents from the forest and pasture treatments where driven by annual variation in temperature and moisture contents whereas turf grass lawn and fallow where dominated by management. The negative correlation of temperature in the pasture and turf grass can be most likely explained by the higher plant productivity during the warmer summer and spring resulting in higher plant $NO_3^-$ and water uptake with increasing temperature which then reduces soil moisture conditions.
Soil mineral N in the SERF forest was generally low and dominated by $NH_4^+$, and while less seasonally variable throughout the year than $NO_3^-$, still responded to rainfall. The overall mineral N reported from temperate eucalypt forest soils was



double the annual SERF average of 17.6 kg N ha$^{-1}$ with up to 38.1 kg N ha$^{-1}$ (Fest et al. 2009; Fest et al. 2015; Livesley et al. 2009). However, the greater NO$_3^-$ proportion in the sandy SERF soil of 3.9 kg N ha$^{-1}$ compared to 0.8 kg N ha$^{-1}$ of temperate sandy forest soils indicates a higher mineral N availability in the subtropics (Livesley et al. 2009). Average NO$_3^-$ contents from other dry sclerophyll forest are even lower with 0.02 kg N ha$^{-1}$ (Fest et al. 2015). The higher N availability is most

likely due to faster soil organic matter turnover in the subtropical climate with higher temperatures in combination with the main annual rainfall. While in temperate summers it is mostly dry during the high temperatures which limits microbial activity, the humid summers in SEQ not only accelerate N turnover but also water and N uptake by plant, which therefore reduces potential N losses. Subtropical rainforests, on the other hand, present with 97.7 kg N ha$^{-1}$ up to 6 times higher mineral N contents than the SERF soil, suggesting a lower N turnover associated with the low net primary productivity

(NPP) of the dry sclerophyll forests (Rowlings et al. 2012). Overall NH$_4^+$:NO$_3^-$ ratios from Australian forests indicate higher NO$_3^-$ availability in subtropical forest soils (3-4) compared to temperate zones (28-125) (Livesley et al. 2009; Rowlings et al. 2012; Fest et al. 2015). These differences in N availability suggest that N cycling in forest soils is mainly regulated by the climate as opposed to soil type and NPP.

Ammonium was the dominant mineral N form in the SERF pasture, similar to the forest and in line with other subtropical

pastures in Australia (Rowlings et al. 2015). The SERF soil reflects the overall minor annual variability of NH$_4^+$ compared to NO$_3^-$ across most climates in Australia. The overall mineral N content at the SERF pasture soil was at the lower end of the reported values from both temperate and subtropical pastures which is most likely explained by the lower clay contents at the site which fixes NH$_4^+$ and higher N inputs by legumes (Livesley et al. 2009; Rowlings et al. 2015). For example, in other extensively used subtropical pastures NH$_4^+$ annual values did not drop below 55 kg N ha$^{-1}$, which is three times higher than

the SERF annual NH$_4^+$ average (Rowlings et al. 2015). While NH$_4^+$ at SERF is comparable to temperate Australian pastures, NO$_3^-$ in the SERF pasture soil is at the lower end (Livesley et al. 2009). This indicates an efficient system from tied up N in organic material to the plant uptake of NO$_3^-$, which supports the hypothesis of an efficient N cycle within well-established land use.

Annual mineral N variations in the SERF turf grass were mainly controlled by the fertilisation events but rapidly fell back to

background levels after each application. The fertilizer mineral N peak was particularly emphasized after the first application, where soil NO$_3^-$ was more than double than after subsequent fertilization events. This is possible due to the undeveloped root system and therefore less N uptake as well as additional plant available N in the added turf grass rolls. These NO$_3^-$ peaks together with irrigation, which is particularly needed during turf grass establishment implies a high N leaching potential which can account for up to 160 kg N ha$^{-1}$ in less than two years in sandy soils (Barton, Wan and Colmer

2006). With the potential of heavy rain events in the subtropics, fertilizer rates and timing needs to be carefully considered to avoid excessive N losses to water ways.

The fallow soil had the highest WFPS content throughout the year due to plant cover removal and therefore no further water uptake by the roots, creating favourable condition for soil mineralisation and losses (Robertson and Groffman 2007). The moist conditions together with the temperatures of the warmer season resulted in accelerated N turnover and without the N





uptake by plants substantial amounts of $NO_3^-$ accumulated in the soil. Despite the fact that mineral N in the fallow soil never dropped back to zero, substantial $NO_3^-$ losses occurred after heavy rain events, not only as gaseous losses but also most likely due to leaching of $NO_3^-$ from the topsoil. The substantial N loss after rain events demonstrates the significant impact of plant cover removal and soil disturbance in peri-urban ecosystems.

**4.2 N₂O fluxes**

The study illustrates that land use change associated with urbanization can significantly alter soil N turnover resulting in elevated soil $N_2O$ emissions and increased N losses from the soil. During the experimental year of this study, autumn was the wettest season and therefore had the highest $N_2O$ emissions from all treatments but with different intensity from the different land use systems. Soil $N_2O$ emissions were significantly different between the investigated land use systems with the

temporal variations in daily $N_2O$ fluxes and primarily controlled by WFPS. However, the linear increase of $N_2O$ emissions with increasing $NO_3^-$ content in the soil of all four treatments suggest higher nitrification than denitrification rates in the SERF soil. The high surface sand content of the Chromosol, combined with the moderate slope, prevents excessive water logging over long periods of time. This reduces $N_2O$ gaseous losses from denitrification, but indicates a high potential for N leaching.

The daily $N_2O$ average of 0.4 g $N_2O$ ha$^{-1}$ d$^{-1}$ from this study's subtropical dry sclerophyll forest is lower than the averages of < 1.2 g $N_2O$ ha$^{-1}$ d$^{-1}$ reported from temperate Australian dry sclerophyll forests (Fest et al. 2009; Livesley et al. 2009). This might be explained by the overall low total C and N and the below average rainfall during the experimental year. Considering the positive correlation of $N_2O$ emissions and $NO_3^-$ content in the soil, it was expected that the higher $NO_3^-$ availability in the SERF forest compared to the temperate dry sclerophyll forest also causes higher $N_2O$ emissions. The low

WFPS which was > 40 % for most of the year inhibited denitrification processes and caused therefore lower $N_2O$ emissions compared to the temperate zones as well as increased $NO_3^-$ uptake during the humid subtropical summer. This efficient N cycling together with the low NPP of the dry sclerophyll forest and low clay content at SERF causes also lower $N_2O$ losses compared to subtropical rainforests (Rowlings et al. 2012). This study supports the general hypothesis that forest soils are minor contributors to the global $N_2O$ budget, although other $N_2O$ emission studies of Australian forest soils provide only a

limited comparison of temporal $N_2O$ variability due to infrequent or short term measurements (Page, Dalal and Raison 2011; Fest et al. 2009; Fest et al. 2015).

The annual $N_2O$ emissions from the SERF pasture are comparable to other reported extensive pastures across Australia (1-2 kg N ha$^{-1}$ y$^{-1}$) but substantially lower than unfertilized pasture in the northern hemisphere (Dalal et al. 2003). Annual emissions from other studies on subtropical Australian pastures reported to be up to 3.4 kg $N_2O$ ha$^{-1}$ y$^{-1}$ and highly inter-

annual variable depending on rainfall (Rowlings et al. 2015). This exceeded the annual $N_2O$ emissions at SERF by nearly 17 times, which may have been limited by the dry year and high sand content.

The first $N_2O$ emission peak after the turf grass's establishment caused the majority of the annual $N_2O$ emissions and was not repeated after two additional fertilization events. This initial $N_2O$ peak can be explained by the underdeveloped root



system and consequently a reduced $NO_3^-$ uptake by the turf grass, which together with the irrigation stimulated nitrification and denitrification and consequential $N_2O$ emissions. The high N demand from the highly productive turf grass later on results in the immediate uptake of mineral N and therefore minor $N_2O$ emissions. The annual $N_2O$ emissions from the SERF's turf grass are more than double the $N_2O$ emissions from extensive Australian pastures reported in the literature

(Dalal et al. 2003). The SERF turf grass lawn emitted about 3.3 times more $N_2O$ on daily average during the experimental year than native pasture from the temperate zones, but only half of the reported values for urban turf grass in the USA which were comparable to intensive agriculture (Kaye et al. 2004). However, compared to Australian intensively managed pastures, $N_2O$ emissions from the SERF turf grass were 50 % lower (Scheer et al. 2011). Differences between reported values and the SERF turf grass are most likely explained by differences in texture and the total N content in the SERF soil being nearly 4

times lower. Reported EFs from temperate pastures also vary substantially between experimental years due to differences in received rainfall (Jones et al. 2005). It could be therefore expected that the SERF's turf grass EF will increase in wetter years. However, in subtropical systems is has been proven that the total amount of annual rainfall received is not as decisive for annual $N_2O$ emissions as rainfall patterns and intensities (Rowlings et al. 2015). These differences between temperate and subtropical N cycling make short term $N_2O$ flux measurements difficult to compare and need further investigation in the

global subtropics.

Significantly higher $NO_3^-$ contents occurred 3 months after plant cover removal in the fallow soil but only during the warm and wet summer season substantial $N_2O$ emissions were observed. The two significant $N_2O$ emission peaks from the fallow were most likely caused by denitrification processes from the accumulated $NO_3^-$ and soil moisture conditions after major rain events. These $N_2O$ emission peaks mirror the $NO_3^-$ decrease from the soil after those rain events but cannot completely

account for it, suggesting that most $NO_3^-$ was leached below 20 cm soil depths or lost via other gases such as $N_2$. All other treatments, including the fertilized turf grass, prevented potential $N_2O$ production in the soil by rapid $NO_3^-$ uptake from plants. Therefore, plant cover removal makes ecosystems undergoing land use change most vulnerable to substantial N losses under humid subtropical climate conditions.

### 4.3 Effect of land use change associated with urbanization

This study determined that land use change associated with urbanization results in substantially elevated $N_2O$ emissions and increases N losses from the soil. The results presented here verify that subtropical $N_2O$ emissions positively correlate to mineral N content in the soil and therefore indicate that land use change increases $N_2O$ emissions from the soil, especially after plant cover removal and establishment of fertilized turf grass lawns. The annual variation in daily $N_2O$ fluxes confirm that despite soil moisture as the strongest climatic parameter influencing $N_2O$ emissions, the individual land use is the main

influence on the soil-atmosphere gas exchange. Extended periods of fallow soil in particular should be avoided during urbanization processes as bare soil is highly vulnerable to N losses due to plant cover removal. Turf grass lawn, as a fertilized and highly managed land cover, leads to significantly changed soil conditions compared to the rural land use types of forest and pasture. However, turf grass lawns in the subtropical climate of SEQ have lower emissions against expectations



based on the high emission findings from temperate zones (Grimm et al. 2008; Tratalos et al. 2007; Kaye et al. 2006). Substantial $N_2O$ emissions were only observed within the first 2 months after turf grass establishment, while over the remaining 10 months only minor fluxes occurred even after further fertilization events. While $N_2O$ emissions from the turf grass were reduced substantially over time, emissions from the fallow increased with time due to more available $NO_3^-$.

Therefore, the $N_2O$ emissions of well-established turf grass lawns need to be considered separately to their production and establishment phase as well as potential N losses from fallow land targeted for the entire duration of land use change, which should be kept as short as possible (Barton, Wan and Colmer 2006; van Delden et al. 2016).

Research from temperate zones suggests a C sequestration potential from the higher productivity of turf grass lawns (Golubiewski 2006; Lorenz and Lal 2009; Raciti et al. 2011). Others argue that the positive effect of C sequestration

however can easily be offset by the high N demand together with irrigation, resulting in increased $N_2O$ emissions and overall nutrient losses caused by management practices like mowing and clipping removal (Conant et al. 2005; Wang et al. 2014). Australian ecosystems with highly weathered soils, however, are generally low in nutrient stocks and often limited in their C sequestration potential (Livesley et al. 2009). The SERF turf grass, however, presented relatively low $N_2O$ emissions when excluding the establishment phase, which implies the potential to balance GHG emissions with C sequestration. A full life

cycle assessment needs to determine if turf grass lawn in the subtropics is increasing or decreasing the GWP of peri-urban environments by balancing C sequestration and GHG emissions, not only from the soil but also through the production, distribution and use of fertilizer, fuel and chemicals (Selhorst and Lal 2011).

## 5 Conclusion

This study provides evidence that land use change associated with urbanization accelerates N turnover and increase $N_2O$

emissions from soils by presenting the first high temporal frequency dataset on peri-urban soils in the subtropics for a full year after land use change. These findings demonstrate that GHG emissions from peri-urban areas should be included into future IPCC climate change scenarios and rural to urban land development guidelines need to be established for GHG emission mitigation. Three main factors need to be considered to target N losses from soils via $NO_3^-$ and $N_2O$ during land use change associated with urbanization: (i) previous land use, (ii) duration of development process, and (iii) new land use

purpose that it is being changed into, i.e. public or private. This dry sclerophyll forest in this study supports the general hypothesis that forest soils are low $N_2O$ emitters, contrary to expectations that the humid subtropical summer conditions would increase emissions compared to temperate forest soils. In the fallow soil the increased N losses via $NO_3^-$ leaching and specifically $N_2O$ emissions may be amplified considering future predictions of rising temperatures and more frequent heavy rain events worldwide. Increased fertilizer application may be required to compensate for these N losses after land use

change to keep peri-urban land such as turf grass highly productive, which subsequently increases GHG emissions. Once established, however, turf grass lawn might offset N losses with potential C sequestration through higher plant productivity.



The outcomes of this study emphasize the potential to mitigate $N_2O$ and $NO_3^-$ losses through altered management during urbanization processes and turf grass establishment.

# 6 Acknowledgements

This study was undertaken at the Samford Ecological Research Facility (SERF) one of the Supersites in the Terrestrial Ecosystem Research Network (TERN). The study was supported by the Institute for Future Environments (IFE) of the Queensland University of Technology (QUT). The data set "Greenhouse gas emissions from peri-urban land use at SERF, SEQ. 2013-2015" can be found online at the $N_2O$ network under http://www.N2O.net.au/knb/metacat/vandelden.3.3/html.

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





**Figure 1: Annual soil $NO_3^-$ (A) and $NH_4^+$ (B) contents variations from forest, pasture, turf grass and fallow averaged across replicates (n = 3) and summed for separate analysed soil depths of 0-10 and 10-20 cm with the climatic conditions (C) for the experimental year 2013/2014 as well as fertilization and irrigation indication for the turf grass treatment.**





**Figure 2: Daily N$_2$O flux averages (max 8 fluxes per day for 3 replicates each) with standard errors (n =3) over the experimental year 2013/2014 for forest (A), pasture (B), turf grass (C) and fallow (D) with the treatment specific water filled pore space (WFPS).**




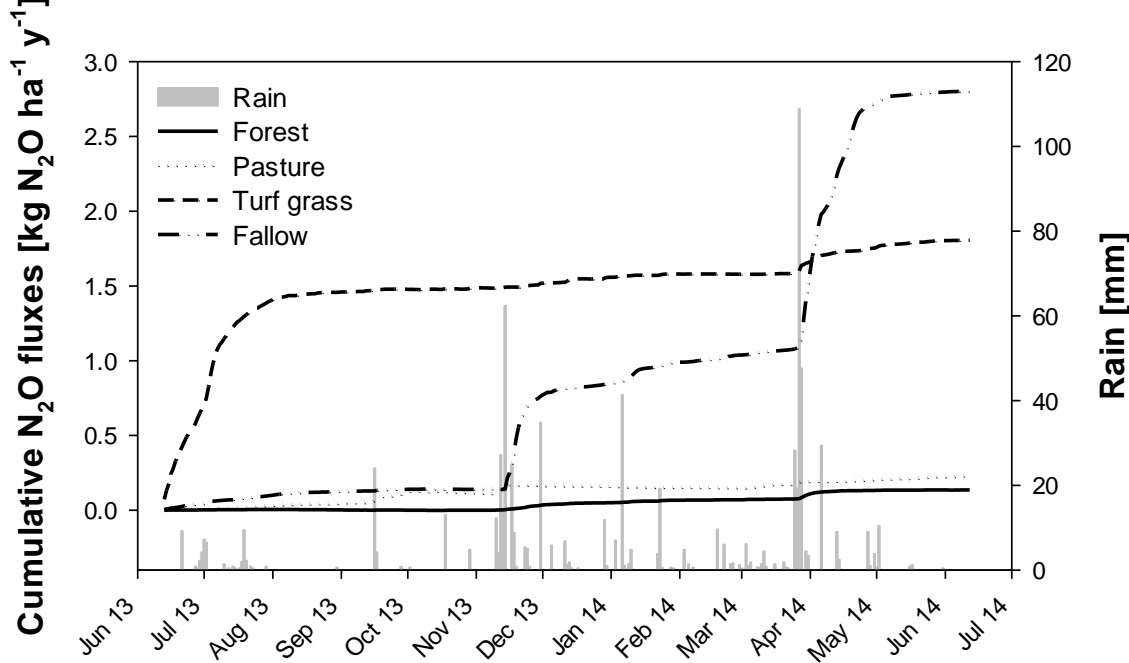

**Figure 3: Cumulative daily N₂O fluxes (n = 3) for forest, pasture, turf grass and fallow with rainfall for the experimental year 2013/2014.**



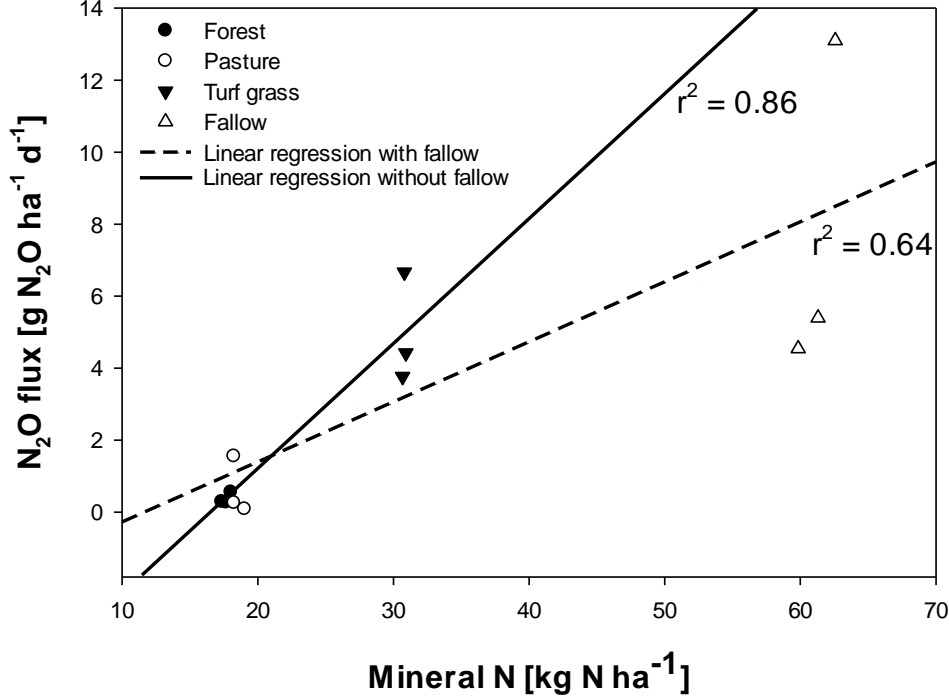

**Figure 4: Linear regression and correlation coefficient $r^2$ for the annual averages of soil mineral N within 20 cm depth (n = 78) and N$_2$O fluxes (n = 1095) for each replicate of forest, pasture, and turf grass as well as with and without fallow.**



**Table 1: SERF site characteristics**

| Parameters | | | | | | | | | | |
|---|---|---|---|---|---|---|---|---|---|---|
| **Longitude** | | | | | 152° 52' 37.3" E | | | | | |
| **Latitude** | | | | | 27° 23' 22.211" S | | | | | |
| **Altitude** | | | | | 60 m | | | | | |
| **Slope** | | | | | 2° | | | | | |
| **Mean annual min temp.** | | | | | 13 °C* | | | | | |
| **Mean annual max temp.** | | | | | 25.6 °C* | | | | | |
| **Mean annual rain** | | | | | 1110 mm* | | | | | |
| **Soil profile** | Horizons** | Depths (cm) | Sand (%) | Silt (%) | Clay (%) | BD (g cm$^{-3}$) | pH | EC (µS) | Total C (%) | Total N (%) |
| **Pasture** | A1 | 0 – 17 | 70 | 24 | 6 | 1.4 | 5.4 | 46 | 1.5 | 0.12 |
| | A2 | 17 – 45 | 74 | 18 | 8 | 1.6 | 6.0 | 10 | 0.9 | 0.07 |
| | B2 | 45 – 92 | 9 | 18 | 73 | 1.8 | 6.1 | 31 | 0.4 | 0.03 |
| **Forest** | A1 | 0 - 20 | 75 | 5 | 20 | 1.4 | 5.5 | 29 | 1.8 | 0.14 |
| | A2 | 20 - 47 | 78 | 5 | 17 | 1.5 | 5.6 | 30 | 1.1 | 0.08 |
| | B2 | 47 - 70 | 44 | 39 | 17 | 1.7 | 5.6 | 30 | 0.2 | 0.02 |

*Long term means by Commonwealth Bureau of Meteorology, Australian Government (BOM)*

*\*\*According to the Australian soil classification*

5   **Table 2: Seasonal and cumulative rain, number of rain events and seasonal and annual averages of minimum and maximum Temperatures of the experimental year**

| | Sum Rain (mm) | Number of rain events* | Avg Temperature (°C) | |
|---|---|---|---|---|
| | | | Min | Max |
| Winter | 51.2 | 0 | 11.5 | 22.6 |
| Spring | 248.2 | 5 | 16.7 | 28.2 |
| Summer | 137.2 | 3 | 20.7 | 30.2 |
| Autumn | 303.2 | 3 | 17.6 | 27.5 |
| | 739.8 | 11 | 16.7 | 27.1 |

*\* Rain event if sum > 10 mm per day*



**Table 3: Annual mineral N averages as NH$_4^+$-N and NO$_3^-$-N in 0-20 cm soil depth, WFPS and daily maximum and average N$_2$O fluxes from all treatments with their cumulative annual fluxes over the experimental year with their standard error**

| | NH$_4^+$-N (kg ha$^{-1}$) | NO$_3^-$-N (kg ha$^{-1}$) | WFPS (%) | Max daily flux (g N$_2$O ha$^{-1}$ d$^{-1}$) | Avg daily flux (g N$_2$O ha$^{-1}$ d$^{-1}$) | Annual flux (kg N$_2$O ha$^{-1}$ y$^{-1}$) |
|---|---|---|---|---|---|---|
| **Forest** | 13.7[a] ± 1.2 | 3.9[a] ± 0.6 | 23[a] | 8.1 | 0.4[a] ± 0.1 | 0.1[a] ± 0.03 |
| **Pasture** | 17.4[b] ± 1.4 | 1.1[b] ± 0.3 | 42[b] | 18.3 | 0.6[a] ± 0.1 | 0.2[a] ± 0.2 |
| **Turf grass** | 21.9[bc] ±2.4 | 8.9[c] ± 2.5 | 43[b] | 83.0 | 4.9[b] ± 0.6 | 1.8[b] ± 0.3 |
| **Fallow** | 26.0[c] ± 1.9 | 35.2[d] ± 5.6 | 55[c] | 123.8 | 7.7[b] ± 1.0 | 2.8[b] ± 1.0 |

[abcd] *different letters indicate significant differences between treatments based on p <0.05*

**Table 4: Spearman's rho correlation coefficient between N$_2$O fluxes and mineral N, WFPS and temperature for each treatment**

| | N$_2$O | | | Mineral N | |
|---|---|---|---|---|---|
| | Mineral N | WFPS | Temperature | WFPS | Temperature |
| **Forest** | -0.39 | 0.61** | 0.40** | 0.07 | -0.17 |
| **Pasture** | 0.49** | 0.30** | -0.46** | 0.39** | -0.41** |
| **Turf grass** | 0.47** | 0.78** | -0.45** | 0.47** | -0.50** |
| **Fallow** | -0.0 | 0.43** | 0.20** | -0.61** | 0.72** |

*** correlation coefficient significant with p < 0.01*