# Peer review of "Urbanization related land use change from forest and pasture into turf grass modifies soil nitrogen cycling and increases N2O emissions"

_Biogeosciences, 2016_

## Referee Comment (RC1) · Anonymous Referee #1 · 18 Jul 2016

This manuscript reports on a work investigating soil N dynamics and nitorus oxide emissions associated with the change in land-use in a transition from forest to peri-urban turf grass establishment. From an experiment perspective I find that the work has been carried out very thoroughly with careful planning and execution giving rise to a solid set of data on soil N, N2O fluxes and key soil parameters. Data from similar ecosystems are rare, and I support these should be made available to the scientific community and do suggest publication of the current work in BG. Meanwhile, I also find that the data interpretations in some places are exaggerated with a need for modifications before publication can be recommended.

As pointed out by the authors the observed N2O emissions from turf grass system is

possibly the result of a system in transition characterized by poor root development during the first months after establishment and thus high N2O emissions due to re-duced N competition by plants. In a longer time perspective, N2O emissions from turf grass systems may remain at low levels throughout the year once the grasses have been established. Meanwhile, I think this need to be emphasized more strongly in the presentation of the results as generalization from the current data should be avoided. Thus, I suggest leaving out the correlation plot (Fig. 4) as in this plot you actually com-pare the established forest and pasture systems with the turf grass supposedly under rapid transition. In this context, I'm also wondering how this correlation analysis can be established on non-transformed data as it is mentioned that data are non-normally distributed. Optionally, the correlation plot may be modified to illustrate partial relation-ships showing the analysis specifically for the initial transition period.

Secondly, the authors conclude that leaching of NO3 took place in the fallow plots. However, this process was not investigated in the current work and although the data may imply water-mediated losses of NO3-N I suggest this statement be modified in the current text.

How were the experimental plots situated in the landscape? E.g. were the random-ized plots separated by strips of pasture, and what was the distance between plots? More details on the gas-flux chambers should be included in the text, e.g. were cham-bers transparent or opaque, how did the chambers open/close. Were fluxes obtained from two pseudo-replicate collars per each replicate plot considered a continuous time-series of measurements, or were they analyzed separately? Please, clarify.

Check citations, if more than two authors only first author + et al. should be mentioned (e.g. Barton, Walm and Colmer, 2006 (P 9)).

In section 4.2 (lines 7 and 14) you refer to a linear increase in N2O emissions with increasing NO3 – where is this shown? Please, clarify.

In section 4.3, line 22 it is concluded that land use change results in increased N losses

from the soil. Is this statement pointing at the N lost as N2O? Please, clarify

Section 5, conclusions – as pointed out above, I think this section needs to be modified according to the nature of the current study. Also, conclusions about turf grass C sequestration cannot be made from this study and should be removed from this section.

---

## Referee Comment (RC2) · Anonymous Referee #2 · 5 Aug 2016

This is a significant contribution to our understanding of the nitrogen cycle in soil-plant systems, and how these are affected by urban development. The manuscript extends work published previously by the team (van Delden et al. (2016)), but as the authors have said the previous work was preliminary data for a shorter period of observation. A much clearer picture is presented in this paper.

However, the key point that comes across is that it is the process of land use change rather than the behavior of established turf grass that affects N2O emissions. I think that the title needs to be changed to reflect this. Also, given the very wide range of possibilities covered by the words 'land use change', a phrase that means very different things to different people, the title should be much more focused, so we know

that the paper addresses changes in N2O emissions as a consequence of turf grass establishment – for lawns, golf courses, amenity grassland etc., and not a permanent change as a consequence of that change.

The behaviour of N depends on the different behaviours of ammonium and nitrate, as explained by the authors. Ammonium is affected by cation exchange, and so is affected by the CEC of the soil and by the mineralogical composition of the soil. While cation exchange is recognized as a process, no data for CEC are presented, and no description of the clay mineralogy of the soil is given. I appreciate that this could be a research project in its own right, but it would help to know what clay minerals are present – perhaps by reference to other published descriptions.

The reason for going on about that is that without knowledge of the CEC (at least) or the mineralogy of the soil, you can't really compare N emissions for different locations, as variation in the clay mineral type could well give rise to different N2O emissions, all other factors being equal. That would be an interesting experiment to conduct.

I have some more minor comments.

1) P2 lines 9 – head or heat? 2) P2 line 31 – what industry? 3) P3 line 11: illustrate, not illustrates 4) P4 line 8 – do you mean 2/3 N by mass or by molar proportion? 5) Section 2.4. You say measurements were made on 'fresh' soil. Was this dried before analysis? Was the soil dried before LECO analysis? I found this section insufficiently detailed to enable me to reproduce the analysis precisely, and it is not clear how the measured analytes were back calculated to the soil under field conditions. Please clear up this ambiguity. 6) P9 line 26 – possibly, instead of possible? 7) Is it possible to calculate a nitrogen balance for these sites?

The references are a bit of a mess, which is surprising these days.

Missing from the bibliography are:

P1 line 23: United Nations 2008 P2 line 8: Grimm 2008 – 2 Grimm et al's are given but

not cited P2 line 30: United Nations 2013 P2 line 34: AGO 2014 P3 line 2: ABS 2012 and Turf Australia 2012 P3 line 20: Moreton Bay Regional Council 2011 P9 line 29 and P12 line 7: it should be Barton et al P10 line 25: it should be Page et al

From the Bibliography the following appear not to be cited (I may have missed some):

Grimm et al – 2 references are given Hart et al 1994 Kaye et al 2006 Lorenz and Lal 2009 Community profile 2011, which is out of order Tratalos et al 2007 Newsletter 2012 again out of order

―――――――――――――――――

---

## Author Comment (AC1) · 1 Sep 2016

The authors would like to thank reviewer 1 for the great feedback. We agree on all of the mentioned suggestions and would prepare the revised manuscript accordingly. The detailed response is as follows:

1. Changing or deleting the correlation plot to avoid generalization of data on establishment phase and the rest of the year.

This recommended modification of correlating the parameters within the establishment phase separately to the rest of the year and changing the data to log transformed values greatly improved the R2 value and results in a valuable partial relationship pre-

sentation. Therefore, we would like to keep but change Figure 4 into the attached figure.

2. Statements about leaching will be reduced and modified to clarify that this is a possibility and not a result of the study.

3. Additional information will be included into the Materials and Method section:

a. Buffer zones of 0.5 m pasture strips between plots

b. Pneumatically operated, clear acrylic glass chambers

c. Fluxes from the two pseudo-replicate frames were analysed continuously as one replicate

4. The bibliography will be updated to make sure the Copernicus output style matches the authors reference list

5. Conclusion section will be modified to clarify the focus of the study: "This study provides evidence that land use change associated with urbanization accelerates N turnover and increase N2O emissions from soils by presenting the first high temporal frequency dataset on peri-urban soils in the subtropics for a full year after land use change. These findings demonstrate that GHG emissions from peri-urban areas should be included into future IPCC climate change scenarios and rural to urban land development guidelines need to be established for GHG emission mitigation. Three main factors need to be considered to target N2O losses from soils during land use change associated with urbanization: (i) previous land use, (ii) duration of development process, and (iii) new land use purpose that it is being changed into, i.e. public or private. This dry sclerophyll forest in this study supports the general hypothesis that forest soils are low N2O emitters, contrary to expectation that the humid subtropical summer conditions would increase emissions compared to temperate forest soils. The accumulation of NO3- in fallow soil increases the potential for N2O emissions and may amplify considering future predictions of rising temperatures and more frequent heavy

rain events worldwide. Increased fertilizer application may be required to compensate for these N losses after land use change to keep peri-urban land, such as turf grass, highly productive, which consequently increases GHG emissions furthermore. The outcomes of this study emphasize the potential to reduce $NO_3^-$ accumulation which increases $N_2O$ emissions through altered management during urbanization processes and turf grass establishment."

[Figure]

[Figure]

**Fig. 1.** Figure 4: Exponential increase of N2O emissions with increasing mineral N content

---

## Author Comment (AC2) · 1 Sep 2016

The authors would like to thank reviewer 2 for the great feedback. We agree on all of the mentioned suggestions and would prepare the revised manuscript accordingly. The detailed response is as follows:

1. Title modification "Urbanization related land use change from forest and pasture into turf grass modifies soil nitrogen cycling and increases N2O emissions"

2. CEC of the soil

CEC analysis will be included into the site description (Table 1) for each soil profile horizon. However, the sandy topsoil showed very low CEC between 0.9 and 4 meq+

100g-1. Even the subsoil (B horizon) with high clay content barely reached 12 meq+ 100g-1. The most occurring clay in the highly weathered soils of Australia is kaolinite with a low CEC of approximately 10 meq+ 100g-1 (Moore et al. 1998).

These CEC results will be included into the text:

In the Materials and Methods section: "The cation exchange capacity (CEC) was determined based on Rayment and Higginson (1992)."

In the Results section: "The CEC of the sandy topsoil is very low, and slightly higher in the A1 compared to the A2 horizon due to the higher soil organic matter as indicated by the total C and N content."

In the Discussion section: "The low CEC of the sandy topsoil highlights the minor nutrient holding capacity of this peri-urban environment."

3. Minor comments will be incorporated as follows:

a. heat

b. turf grass industry

c. illustrate

d. two-thirds of the N content

e. "NH4+ and NO3- were extracted from the soil using a 1:5 KCl solution with 20 g of fresh soil with additional soil moisture determination at 105°C to identify the dry soil weight for the mineral N calculation as described by Carter and Gregorich (2007)." And "Total C and N content of air dried soil and plant material was determined by dry combustion (CNS-2000, LECO Corporation, St. Joseph, MI, USA) from ground samples."

f. possibly

g. We agree on the advantage of a N balance for these systems. However, as reviewer

1 highlighted, this study did not quantify nitrate leaching, which seems to be one of the major N losses in these systems. Therefore, a rough calculated N balance based solely on fertilizer input and N2O emissions would not be representative but will be subject for further research.

4. The bibliography will be updated to make sure the Copernicus output style matches the authors reference list

---

## Author Response (AR1)

Revisions on the manuscript: "Land use change associated with urbanization modifies soil nitrogen cycling and increases $N_2O$ emissions"

Editor:

5  Dear Authors

thank you very much for your final responses to the reviews.

I have a few comments:

1. please provide a revision with track changes.

2. in your responses it will make it easier for us , if you copied the reviewer comment and then let this follow by your

10  response.

3. In the revised Figure 4 you speak of an exponential relationship. Could you try log (y) linear x scale, then the exponential

nature would figure as a line.

With kind regards,

Andreas Ibrom

The authors would like to thank Andreas Ibrom for the comments. Figure 4 was updated accordingly, which improved the $R^2$ even further. This suggestion was much appreciated.

[Figure]

20  Figure 1: Linear relationship of log transformed $N_2O$ emissions with mineral N content within 20 cm soil depth for each replicate of forest, pasture, turf grass and fallow land use during the establishment phase (A) and the rest of the year (B), with the coefficient of determination $R^2$.

The authors would like to thank both reviewers for their great feedback. We agree on all of the mentioned suggestions and prepared the revised manuscript accordingly. The detailed response is as follows:

Referee 1

5  This manuscript reports on a work investigating soil N dynamics and nitorus oxide emissions associated with the change in land-use in a transition from forest to peri-urban turf grass establishment. From an experiment perspective I find that the work has been carried out very thoroughly with careful planning and execution giving rise to a solid set of data on soil N, N2O fluxes and key soil parameters. Data from similar ecosystems are rare, and I support these should be made available to the scientific community and do suggest publication of the current work in BG. Meanwhile, I also find that the data
10  interpretations in some places are exaggerated with a need for modifications before publication can be recommended. As pointed out by the authors the observed N2O emissions from turf grass system is possibly the result of a system in transition characterized by poor root development during the first months after establishment and thus high N2O emissions due to reduced N competition by plants. In a longer time perspective, N2O emissions from turf grass systems may remain at low levels throughout the year once the grasses have been established. Meanwhile, I think this need to be emphasized more
15  strongly in the presentation of the results as generalization from the current data should be avoided.

Thus, I suggest leaving out the correlation plot (Fig. 4) as in this plot you actually compare the established forest and pasture systems with the turf grass supposedly under rapid transition. In this context, I'm also wondering how this correlation analysis can be established on non-transformed data as it is mentioned that data are non-normally distributed. Optionally, the correlation plot may be modified to illustrate partial relationships showing the analysis specifically for the initial transition
20  period.

This recommended modification of correlating the parameters within the establishment phase separately to the rest of the year and changing the $N_2O$ data to log transformed values greatly improved the $R^2$ value and results in a valuable partial relationship presentation. This suggestion was much appreciated. The figure was furthermore updated according to the editor's comments as mentioned above.

Secondly, the authors conclude that leaching of NO3 took place in the fallow plots. However, this process was not investigated in the current work and although the data may imply water-mediated losses of NO3-N I suggest this statement be modified in the current text.

Statements about leaching was reduced and modified to clarify the focus of the study:

30  Section 4.1, page 8, line 29:"Nutrient mineralisation is often faster in sandy soils but the rapid infiltration and low nutrient holding capacity of the A horizon of the Chromosol decreases the highly mobile $NO_3^-$ content substantially after heavy rain events. This amount of $NO_3^-$ is not only lost for plant uptake but can pollute groundwater and open waterways resulting in eutrophication."
Section 4.1, page 9, line 28:" These $NO_3^-$ peaks together with irrigation, which is particularly needed during turf grass
35  establishment, implies a high N leaching potential in other Australian sandy soils of up to 80 kg N ha$^{-1}$ y$^{-1}$ (Barton et al. 2006). With the high potential of heavy rain events in the subtropics, fertilizer rates and timing needs to be considered carefully to avoid excessive N losses in form of $NO_3^-$ displacement."

Section 4.1, page 10, line 3:" Despite the fact that mineral N in the fallow soil never dropped back to zero, substantial amounts of $NO_3$ were lost from the topsoil after heavy rain events, not only as $N_2O$ emissions but also through $NO_3^-$ displacement into deeper soil layers."

Section 4.2, page 10, line 14:" The high surface sand content of the Chromosol, combined with the moderate slope, prevents excessive water logging over long periods of time, which limits $N_2O$ gaseous losses from denitrification in saturated soil conditions."

Section 4.3 from line 28:" The results presented here verify that subtropical $N_2O$ emissions positively correlate to mineral N content in the soil and therefore indicate that land use change increases $N_2O$ emissions from the soil, especially after plant cover removal and establishment of fertilized turf grass lawn."

How were the experimental plots situated in the landscape? E.g. were the randomized plots separated by strips of pasture, and what was the distance between plots? More details on the gas-flux chambers should be included in the text, e.g. were chambers transparent or opaque, how did the chambers open/close. Were fluxes obtained from two pseudo-replicate collars per each replicate plot considered a continuous timeseries of measurements, or were they analyzed separately? Please, clarify.

Additional information included into the Materials and Method section:

    a.   Buffer zones of 0.5 m pasture strips between plots
    b.   Pneumatically operated, clear acrylic glass chambers
    c.   Fluxes from the two pseudo-replicate frames were analysed continuously as one replicated treatment

Check citations, if more than two authors only first author + et al. should be mentioned (e.g. Barton, Walm and Colmer, 2006 (P 9)).

The bibliography will be updated to make sure the Copernicus output style matches the author's reference list.

In section 4.2 (lines 7 and 14) you refer to a linear increase in N2O emissions with increasing NO3 – where is this shown? Please, clarify.

Clarified into: "However, the linear increase of $N_2O$ emissions with increasing $NO_3^-$ content in the soil may be the result of higher denitrification than nitrification rates in the SERF soil."

And sentence added into results section to clarify the linear relationship shown in the updated Figure 4: "However, the linear regression shown in Figure 4 identified a clear increase of $N_2O$ emissions with increasing annual mineral N contents for all treatments during the establishment phase as well as during the rest of the year."

In section 4.3, line 22 it is concluded that land use change results in increased N losses from the soil. Is this statement pointing at the N lost as N2O? Please, clarify

Clarified into: "This study determined that urbanization related land use change results in an accumulation of $NO_3^-$ in fallow topsoil and elevated $N_2O$ emissions, mainly after heavy rain events."

Section 5, conclusions – as pointed out above, I think this section needs to be modified according to the nature of the current study. Also, conclusions about turf grass C sequestration cannot be made from this study and should be removed from this section.

Conclusion section will be modified to clarify the focus of the study:

5 "This study provides evidence that land use change associated with urbanization accelerates N turnover and increase $N_2O$ emissions from soils by presenting the first high temporal frequency dataset on peri-urban soils in the subtropics for a full year after land use change. These findings demonstrate that GHG emissions from peri-urban areas should be included into future IPCC climate change scenarios and rural to urban land development guidelines need to be established for GHG emission mitigation. Three main factors need to be considered to target $N_2O$ losses from soils during land use change 10 associated with urbanization: (i) previous land use, (ii) duration of development process, and (iii) new land use purpose that it is being changed into, i.e. public or private. The dry sclerophyll forest in this study supports the general hypothesis that forest soils are low $N_2O$ emitters, contrary to expectation that the humid subtropical summer conditions would increase emissions compared to temperate forest soils. The accumulation of $NO_3^-$ in fallow soil increases the potential for $N_2O$ emissions and may amplify considering future predictions of rising temperatures and more frequent heavy rain events 15 worldwide. Increased fertilizer application may be required to compensate for these N losses after land use change to keep land uses, such as turf grass, highly productive while altering N cycling in peri-urban environments. The outcomes of this study highlight the substantial $NO_3^-$ accumulation in soils during land use change which consequently increases $N_2O$ emissions and should be accounted for in global climate forecasts as urbanization processes are predicted to increase worldwide with increasing population growth."

Referee 2

This is a significant contribution to our understanding of the nitrogen cycle in soil-plant systems, and how these are affected by urban development. The manuscript extends work published previously by the team (van Delden et al. (2016)), but as the 25 authors have said the previous work was preliminary data for a shorter period of observation. A much clearer picture is presented in this paper. However, the key point that comes across is that it is the process of land use change rather than the behavior of established turf grass that affects N2O emissions. I think that the title needs to be changed to reflect this. Also, given the very wide range of possibilities covered by the words 'land use change', a phrase that means very different things to different people, the title should be much more focused, so we know that the paper addresses changes in N2O emissions 30 as a consequence of turf grass establishment – for lawns, golf courses, amenity grassland etc., and not a permanent change as a consequence of that change.

The authors agree and suggest a title change into:

"Urbanization related land use change from forest and pasture into turf grass modifies soil nitrogen cycling and increases $N_2O$ emissions"

35 The behaviour of N depends on the different behaviours of ammonium and nitrate, as explained by the authors. Ammonium is affected by cation exchange, and so is affected by the CEC of the soil and by the mineralogical composition of the soil. While cation exchange is recognized as a process, no data for CEC are presented, and no description of the clay mineralogy of the soil is given. I appreciate that this could be a research project in its own right, but it would help to know what clay

minerals are present – perhaps by reference to other published descriptions. The reason for going on about that is that without knowledge of the CEC (at least) or the mineralogy of the soil, you can't really compare N emissions for different locations, as variation in the clay mineral type could well give rise to different N2O emissions, all other factors being equal. That would be an interesting experiment to conduct.

5    The authors appreciate the suggested importance of the specific site characteristic CEC and include CEC analysis into the site description (Table 1) for each soil profile horizon.
However, the sandy topsoil showed very low CEC between 0.9 and 4 $meq^+$ $100g^{-1}$, which suggests a minor effect of CEC on the $NH_4^+$ dynamics. Even the subsoil (B horizon) with high clay content barely reached 12 $meq^+$ $100g^{-1}$. The most occurring clay in the highly weathered soils of Australia is kaolinite with a low CEC of approximately 10 $meq^+$ $100g^{-1}$ (Moore et al.
10   1998).

These CEC results are included into the text:

In section 2.4: "The cation exchange capacity (CEC) was determined based on Rayment and Higginson (1992)."

In section 3.1: "The CEC of the sandy topsoil is very low, and slightly higher in the A1 compared to the A2 horizon due to the higher soil organic matter as indicated by the total C and N content."

15   In section 4.1, page 8, line 24: "Nutrient mineralisation is often faster in sandy soils but the rapid infiltration and low nutrient holding capacity of the A horizon of the Chromosol decreases the highly mobile $NO_3^-$ content substantially after heavy rain events."

In section 4.1, page 10, line 11: "The low CEC of the sandy topsoil highlights the minor nutrient holding capacity of this peri-urban environment."

20   I have some more minor comments.

Minor comments are addressed as followed:

1) P2 lines 9 – head or heat?

Heat

2) P2 line 31 – what industry?

25   Turf grass industry

3) P3 line 11: illustrate, not illustrates

Illustrate

4) P4 line 8 – do you mean 2/3 N by mass or by molar proportion?

Two-thirds of the N content by mass

30   5) Section 2.4. You say measurements were made on 'fresh' soil. Was this dried before analysis? Was the soil dried before LECO analysis? I found this section insufficiently detailed to enable me to reproduce the analysis precisely, and it is not clear how the measured analytes were back calculated to the soil under field conditions. Please clear up this ambiguity.

"Mineral N in form of $NH_4^+$ and $NO_3^-$ were extracted from the soil using a 1:5 KCl solution with 20 g of fresh soil with additional soil moisture determination at 105°C to identify the dry soil weight for the mineral N calculation as described by
35   Carter and Gregorich (2007))."

"Total C and N content of air dried soil and plant material was determined by dry combustion (CNS-2000, LECO Corporation, St. Joseph, MI, USA) from ground samples."

6) P9 line 26 – possibly, instead of possible?

Possibly

7) Is it possible to calculate a nitrogen balance for these sites?

We agree on the advantage of a N balance for these systems. However, as reviewer 1 highlighted, this study did not quantify nitrate leaching, which seems to be one of the major N losses in these systems. Therefore, a rough calculated N balance based solely on fertilizer input and $N_2O$ emissions would not be representative but will be subject for further research.

The references are a bit of a mess, which is surprising these days. Missing from the bibliography are: P1 line 23: United Nations 2008 P2 line 8: Grimm 2008 – 2 Grimm et al's are given but not cited P2 line 30: United Nations 2013 P2 line 34: AGO 2014 P3 line 2: ABS 2012 and Turf Australia 2012 P3 line 20: Moreton Bay Regional Council 2011 P9 line 29 and P12 line 7: it should be Barton et al P10 line 25: it should be Page et al From the Bibliography the following appear not to be cited (I may have missed some): Grimm et al – 2 references are given Hart et al 1994 Kaye et al 2006 Lorenz and Lal 2009 Community profile 2011, which is out of order Tratalos et al 2007 Newsletter 2012 again out of order

We apologise for the mess, the bibliography will be updated to make sure the Copernicus output style matches the author's reference list.

Please find the marked-up manuscript version below:

[revised manuscript text omitted]

---

## Referee Report (RR1)

Urbanization related land use change from forest and pasture into turf grass modifies soil nitrogen cycling and increases $N_2O$ emissions
Thanks for making the recommended corrections – the paper reads well and is a good story that makes interesting reading.

As an old pedant, I have a very few minor corrections:

Page 9 line 3: you say 'where' twice, when you mean 'were'

Page 12 line 6: it should be 'confirms' not 'confirm'

Please proof read carefully in case there are other minor errors of this type.

Reference list: the format of the author names is inconsistent.  Some use initials, and others give first names in full.  Please make sure that all are consistent with the style required for Biogeosciences Discussions.

---

## Author Response (AR2)

Referee #1, Report from 02 Oct 2016

Urbanization related land use change from forest and pasture into turf grass modifies soil nitrogen cycling and increases N2O emissions Biogeosciences Discussions

Thanks for making the recommended corrections – the paper reads well and is a good story that makes interesting reading. As an old pedant, I have a very few minor corrections:

Page 9 line 3: you say 'where' twice, when you mean 'were'

Changed

Page 12 line 6: it should be 'confirms' not 'confirm'

Changed

Please proof read carefully in case there are other minor errors of this type.

The manuscript was proof read again

Reference list: the format of the author names is inconsistent. Some use initials, and others give first names in full. Please make sure that all are consistent with the style required for Biogeosciences Discussions.

Changes made accordingly to match the Biogeoscience regulations